# A new motor synergy that serves the needs of oculomotor and eye lid systems while keeping the downtime of vision minimal

Mohammad Farhan Khazali[1,2]*, Joern K Pomper[1,2], Aleksandra Smilgin[1,2], Friedemann Bunjes[1,2], Peter Thier[1,2]*

[1]Department of Cognitive Neurology, University of Tübingen, Tübingen, Germany; [2]Hertie Institute for Clinical Brain Research, University of Tübingen, Tübingen, Germany

**Abstract** The purpose of blinks is to keep the eyes hydrated and to protect them. Blinks are rarely noticed by the subject as blink-induced alterations of visual input are blanked out without jeopardizing the perception of visual continuity, features blinks share with saccades. Although not perceived, the blink-induced disconnection from the visual environment leads to a loss of information. Therefore there is critical need to minimize it. Here we demonstrate evidence for a new type of eye movement serving a distinct oculomotor demand, namely the resetting of eye torsion, likewise inevitably causing a loss of visual information. By integrating this eye movement into blinks, the inevitable down times of vision associated with each of the two behaviors are synchronized and the overall downtime minimized.

*For correspondence: mohammad.khazali@student.uni-tuebingen.de (MFK); thier@uni-tuebingen.de (PT)

**Competing interests:** The authors declare that no competing interests exist.

## Introduction

We normally blink around 15 to 20 times per minute to hydrate the eyes and to protect them from potentially harmful agents (*Evinger, 1995*; *Evinger et al., 1991*). Although normal visual input is blanked out for ~300 milliseconds during blinks, we rarely perceive these gaps due to what is called blink suppression (*Bristow et al., 2005*; *Volkmann et al., 1980*; *Volkmann, 1986*; *Manning et al., 1983*), the attenuation of visual sensitivity during blinks. This mechanism is useful to eliminate the visual consequences of eye and lid movements during blinks (*Ridder and Tomlinson, 1997*; *Volkmann et al., 1968*). But it comes with the cost of losing the connection to the visual world for the duration of the blink (*VanderWerf et al., 2003*; *Bour et al., 2000*). Without compensation this loss would lead to a discontinuous visual perception. However, previous experiments in humans have shown that the visual system is able to ensure an illusion of continuity by bridging the perceptual gaps (*Bristow et al., 2005*; *Shultz et al., 2011*). Moreover, in cases quick behavioral responses may be demanded, the blink frequency may be temporarily reduced in order to provide more time for the processing of behaviorally relevant visual information. For example during reading, we blink only 4–5 instead of 15–20 times per minute (*Shultz et al., 2011*; *Bentivoglio et al., 1997*). Another mechanism, reducing the down time of the visual system is the synchronization of blinks with saccades, the latter likewise blanking out vision due to saccadic suppression (*Fogarty and Stern, 1989*; *Zee et al., 1983*; *Evinger et al., 1991*, *1984*).

Not only voluntary saccades interfere with vision but also the fast resetting movements that are part of oculomotor reflexes that stabilize the visual background. One of these reflexes is the torsional optokinetic nystagmus (tOKN) (*Farooq et al., 2004*). It consists of a slow phase in which the

**eLife digest** Blinking lubricates the surface of the eye and prevents it from becoming too dry. People blink around 20 times per minute, but we rarely notice the gaps in our vision because the brain becomes less sensitive to visual input during each blink.

Eye movements can also hinder our vision. For example, our vision could easily become blurred when we redirect our gaze to a new target. However, like during blinking, the brain blocks visual input while these movements – known as saccades – take place. Because saccades and blinking both temporarily disrupt vision, the brain synchronizes the two, which keeps this downtime to a minimum.

Involuntary eye movements can also disrupt our vision. We perceive an object most clearly when its image falls on the very center of the back of the eye. If the image drifts away from this central position, our eyes make small involuntary 'resetting' movements to counteract the drifting. Given that these movements would compromise our vision much like saccades do, Khazali et al. wondered whether the brain might also synchronize them with blinks as well.

Khazali et al. tracked the eye movements of people staring straight ahead at a red dot in an otherwise dark room. This revealed that correcting eye movements did indeed occur at the same time as blinks, but unexpectedly the movements were different from those that had been recorded in previous studies. Further experiments then confirmed that the blinks themselves triggered this new type of corrective movement, rather than the loss of vision that resulted from the blinks. Khazali et al. propose that by synchronizing the resetting movements with blinks, the brain ensures that the eyes are damp wiped and restored to their optimal positions while minimizing visual disturbance. The next challenge is to identify the circuit of neurons in the brain that underlies these eye movements and to clarify how this circuit interacts with the one that controls blinking.

eyes pursue the rotation of the visual stimulus in order to reduce the slip of its retinal image, followed by a torsional fast phase in the opposite direction. The fast phase, generally considered to rely on the brainstem premotor circuitry for saccades (*Tozzi et al., 2007*), ensures that the eyes stay within the mechanical limits of the plant. In view of the tight interaction of voluntary saccades and blinks we expected that also the involuntary fast phases of the tOKN might be synchronized with blinks in order to minimize the overall down time of the visual system. Here we report that the over-arching goal to optimize visual perception is actually met in a surprisingly different manner. Rather than combining fast phases with blinks, the oculomotor system deploys a novel type of resetting movement which is qualitatively different from fast phases.

## Results

### Blinks reset the torsional eye position

Experiments were carried out in a dark room with subjects sitting in front of a tangent screen on which visual stimuli were presented. During the whole experiment they had to fixate a red dot located straight ahead. Each experimental block consisted of four phases (see *Figure 1*, and Materials and methods).

Eye position was recorded by 3-D video-oculography (for details, see Materials and methods). The mean torsional eye position during the first phase (fixation in darkness I), when only the central fixation dot was visible, was taken as reference for eye torsion (referred to as zero torsion from now on). During the 'fixation in darkness' phase I, which lasted for 180 s, the eye torsion fluctuated around zero by ± 2 deg (mean ± std, by averaging the individual means of subjects, calculated as the average of the eye position for the 180 s). In the second phase (optic flow I), a large field visual background (108 deg x 90 deg) appeared on the screen that rotated around the fixation dot at 30 deg/s clockwise, or, alternatively counterclockwise. When subjects were retested on other days the rotation direction experienced was the respective other one (see Materials and methods).

This torsional optic flow stimulus was presented for 270 s. The eyes followed the torsional flow with a mean velocity of ~2.95 ± 2.9 deg/s (mean ± std across subjects). This following movement, the slow phase of the torsional optokinetic nystagmus (tOKN), was time and again interrupted by fast

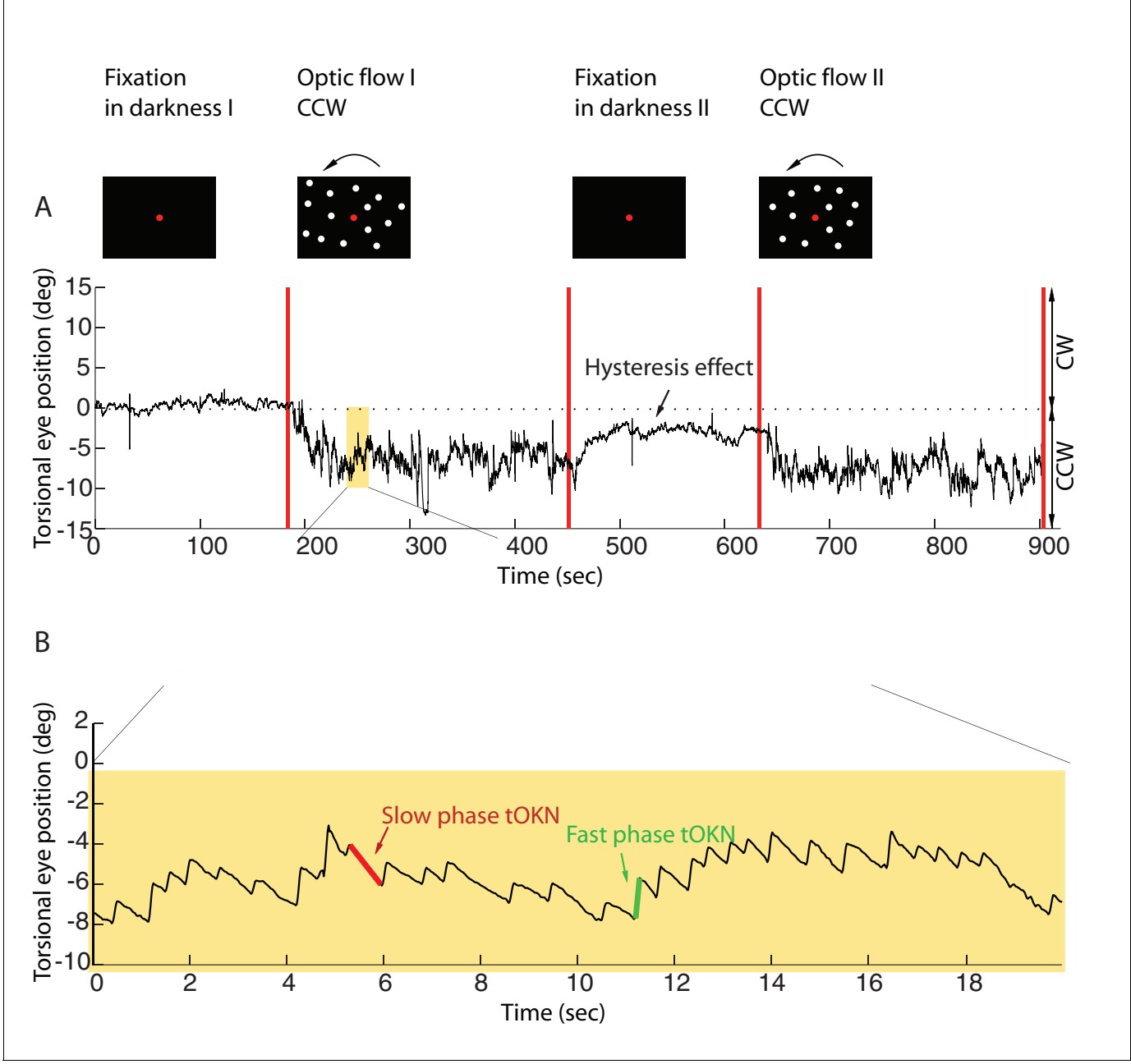

**Figure 1.** Torsional eye movements evoked by an optokinetic stimulus. (**A**) An example of the experimental block depicting the stimulus in the upper row and the associated torsional eye position beneath. A block consisted of four phases. 'Fixation in darkness' phase I was a fixation period (start from zero), followed by a optic flow phase I phase in which a large field visual stimulus rotating counterclockwise was presented on the monitor (start marked with a red line). The subsequent 'fixation in darkness' phase II and 'optic flow' phase II were identical to 'fixation in darkness' phase I and 'optic flow' phase I, respectively. Note the build-up of eye torsion during the stimulus presentation. (**B**) Enlarged torsional eye position from (**A**) showing torsional optokinetic nystagmus consisting of fast phases in green towards the zero torsional eye position and slow phases in red which follow the direction of the rotating stimulus. Notice that the eye torsional position did not return completely back to zero, i.e. it showed a hysteresis effect.

movements in the opposite direction, corresponding to the fast phases of the tOKN, which aim to move the eyes back towards zero torsion. This resetting was, however, imperfect as reflected by a gradual build-up in the direction of the optic flow until a plateau of ~3–8 deg was reached. This plateau varied between subjects and experimental blocks. This build-up is a consequence of a smaller cumulative amplitude of fast phases as compared to the cumulative amplitude of slow phases.

When the optic flow phase had ended, a fixation period of 180 s duration in total darkness followed (fixation in darkness II). During this time the overall eye torsion slowly drifted back towards zero torsion, yet, usually without fully reaching it. Finally, the fourth phase (optic flow II), which replicated the stimulus conditions of phase 2, completed the experimental block.

Blinks could easily be detected and distinguished from saccades in the video-oculographic records based on their characteristic signature (see Materials and methods). During the 'optic flow' phases I and II, once the eyes showed an optokinetic nystagmus and had gradually reached their respective plateau of up to 8 deg, we observed a change in the eye position towards zero torsion associated with each blink. We will refer to this eye movement as blink-associated resetting movement, BARM (see *Figure 2B,E*). The amplitudes of these BARMs were in the order of one degree (1.34 ± 1.3 deg, mean ± std across subjects). In the 'fixation in darkness' phase I, when only the fixation dot was present and deviations from zero were small, BARMs were also seen, but they were less frequent, more variable in direction and smaller (0.6 ± 0.89 deg, mean ± std, see *Figure 2A,D*). In the 'fixation in darkness' phase II, however, in which the visual conditions were not different from the 'fixation in darkness' phase I, but the average eye position had not yet fully returned to zero torsion, BARMs were much more frequent and of larger amplitude (0.68 ± 0.79 deg mean ± std, p=0.02, fixation in darkness phase II vs. fixation in darkness phase I, see *Figure 2C,F* for one of the subjects; for the other subjects see *Figure 2—figure supplement 1*). This suggests that the probability and the amplitude of BARMs are determined by the deviation of the eyes from zero torsion, no matter if an

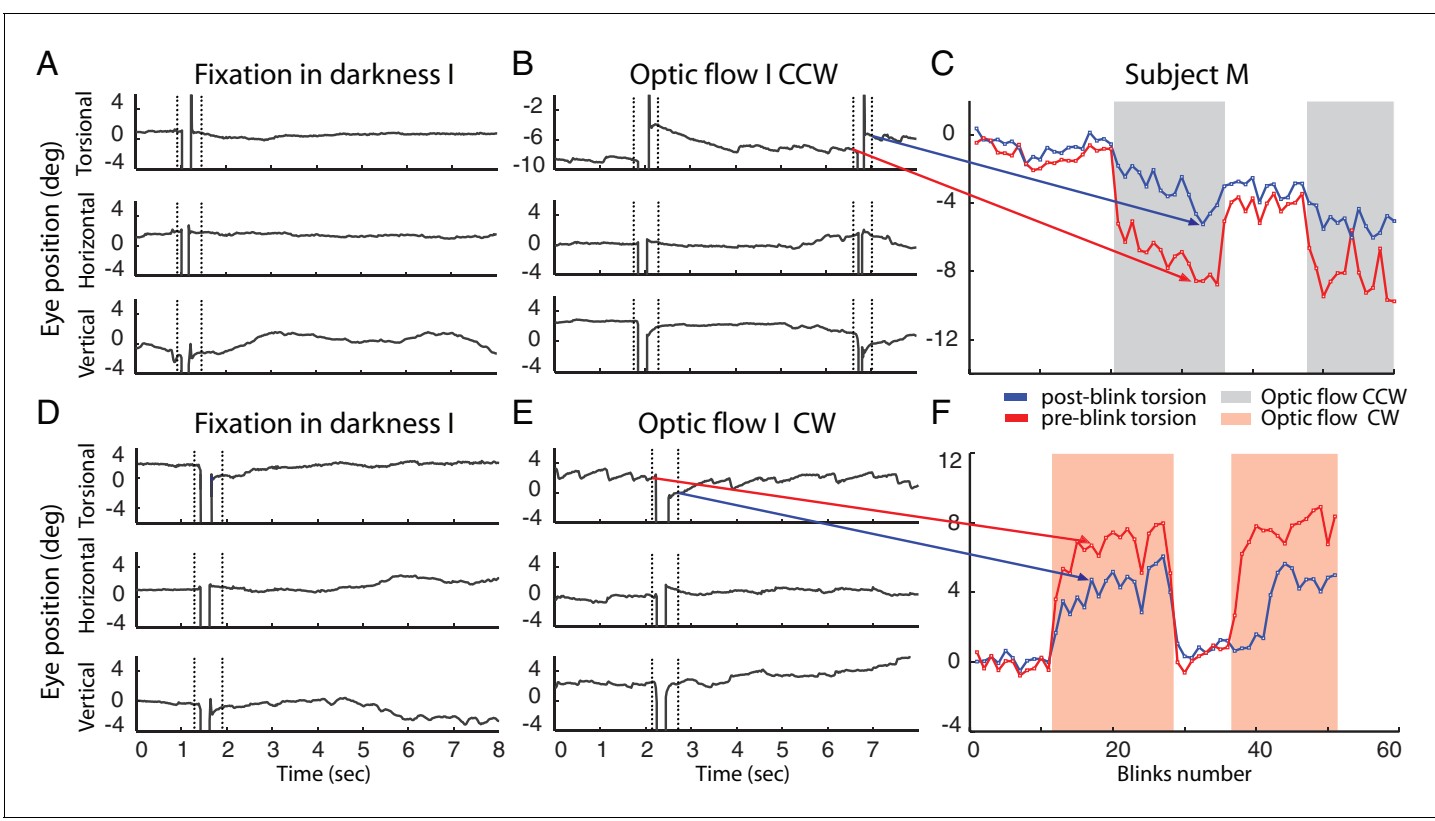

**Figure 2.** Example of blinks during fixation in darkness (**A**, **D**) and during the presentation of the optic flow pattern (**B**, **E**). In (**B**) the optic flow is counterclockwise, in (**E**) clockwise. Dotted lines demarcate the period around the blink-associated artifact of the eye position record, which was not considered for analysis (see Materials and methods). Note the torsional eye position shift towards zero after every blink during the presentation of optic flow stimulus, and less consistently in darkness. (**C** and **F**) are comparing the pre- and post-blink torsional eye position for all blinks observed in two blocks examples of subject M.

The following figure supplement is available for figure 2:

**Figure supplement 1.** Torsional eye position before and after blinking for one experimental block per optic flow direction.

optokinetic stimulus is present or not. In accordance with this view, we found the amplitude of BARMs to be linearly correlated with the size of the pre-blink torsional eye position, which is the deviation from zero torsion (see *Figure 3A*). In contrast, the amplitude of BARMs was not related to the slow phase preceding the blink, since the amplitude of the slow phase, defined by the difference of the torsional eye position between the onset of the slow phase and the onset of the blink, was not correlated with the amplitude of BARMs (see *Figure 3B*, for the other subjects see *Figure 3— figure supplement 1*). In other words, the absolute torsional eye position matters but the latest history of the slow phase does not. Although the correlation between the amplitude of BARMs and the pre-blink eye position was strongest for the 'optic flow' phase I and II (see *Table 1*), in which the optokinetic stimulus was present and the deviation from zero was largest, we found similar relationships also for the 'fixation in darkness' phase I and phase II. This supports the view that BARMs are not restricted to optic flow stimuli but that they are a ubiquitous phenomenon. In accordance with the correlation analysis, also the slope of the regression of the BARM amplitude as a function of the pre-blink torsional eye position was largest for the 'optic flow' phase I and II, intermediate for the 'fixation in darkness' phase II and smallest for the 'fixation in darkness' phase I. In order to elucidate if theses experimental phase differences were a consequence of different ranges of pre-blink torsional eye positions in the various phases, the regressions and the correlations were recalculated for exactly the same, constrained range of pre-blink torsional eye positions (−2 deg to 2 deg). *Table 1* shows that the differences between phases in terms of the quality of fits and the slopes of the regression lines stayed. This indicates that the BARM amplitude is not only dependent on the pre-blink torsional eye position preceding a single BARM but additionally influenced by the average torsional eye position of a phase. It suggests a gain modulation of the linear relation between pre-blink torsional eye position and BARM amplitude according to superordinate requirements.

Are BARMs, in fact, fast phases coinciding with blinks? The occurrence of BARMs during the 'fixation in darkness' phases I and II argues against this objection. In these phases there was no optokinetic stimulus present that could give rise to a fast phase optokinetic response. Moreover, the fixation in darkness period I did not follow a preceding exposure to an optokinetic stimulus, excluding fast phases as hangovers of earlier optokinetic stimulation. Further evidence for a qualitative difference between BARMs and fast phases comes from an analysis that we performed for fast phases in analogy to the one for BARMs described before. In the case of fast phases, the preceding slow phase amplitude, i.e. the difference between the onset and the offset of the slow phase explains a large part of the amplitude of the following fast phase (r = −0.35 ± 0.1 mean ± std. across subjects, *Figure 3* and *Figure 3—figure supplement 1*, mean slope of 0.33). On the other hand, the torsional eye position at the onset of the fast phase has little if any influence on the size of the fast phase amplitude (correlations are only weak and inconsistent: significant correlation in 3 out of 5 subjects, r = −0.18 ± 0.09 mean ± std. across subjects, see *Figure 3* and *Figure 3—figure supplement 1*). This is very different from BARMs, in which – as said earlier - it is the pre-blink torsional eye position which determines the size of the BARM. Moreover, the amplitude of the fast phases are dependent on the optic flow direction. This is why they form two separated clusters in plots of fast phase amplitude as function of the direction of the optic flow. Fitting these two clusters requires separate linear regressions yielding different regression slopes and offsets in comparison with fitting them as one pool (see Materials and methods). On the other hand, BARMs form a continuous distribution with similar regression slopes and offsets independent of the direction of the optic flow. This was confirmed by having almost no difference between the regression slopes produced by fitting blinks as two separated pools, depending on the optic flow direction, and by fitting them as one (see details in *Figure 3*, *Figure 3—figure supplement 1* legends and Materials and methods). We conclude that BARMs and fast phases are functionally two distinct entities (see *Figure 3C and F*).

## Blink-associated resetting of eye torsion is realized by a specific type of eye movement

Although functionally distinct from fast phases, BARMs could nevertheless be a kinematic product of the saccade machinery, simply working under a different control command (torsional eye position instead of slow phase amplitude). Alternatively, they could be modifications of blink-associated eye movements that occur during fixation and that have been described in studies of visual fixation in the absence of optic flow. These movements are directed nasally, downwards and extorsionally and

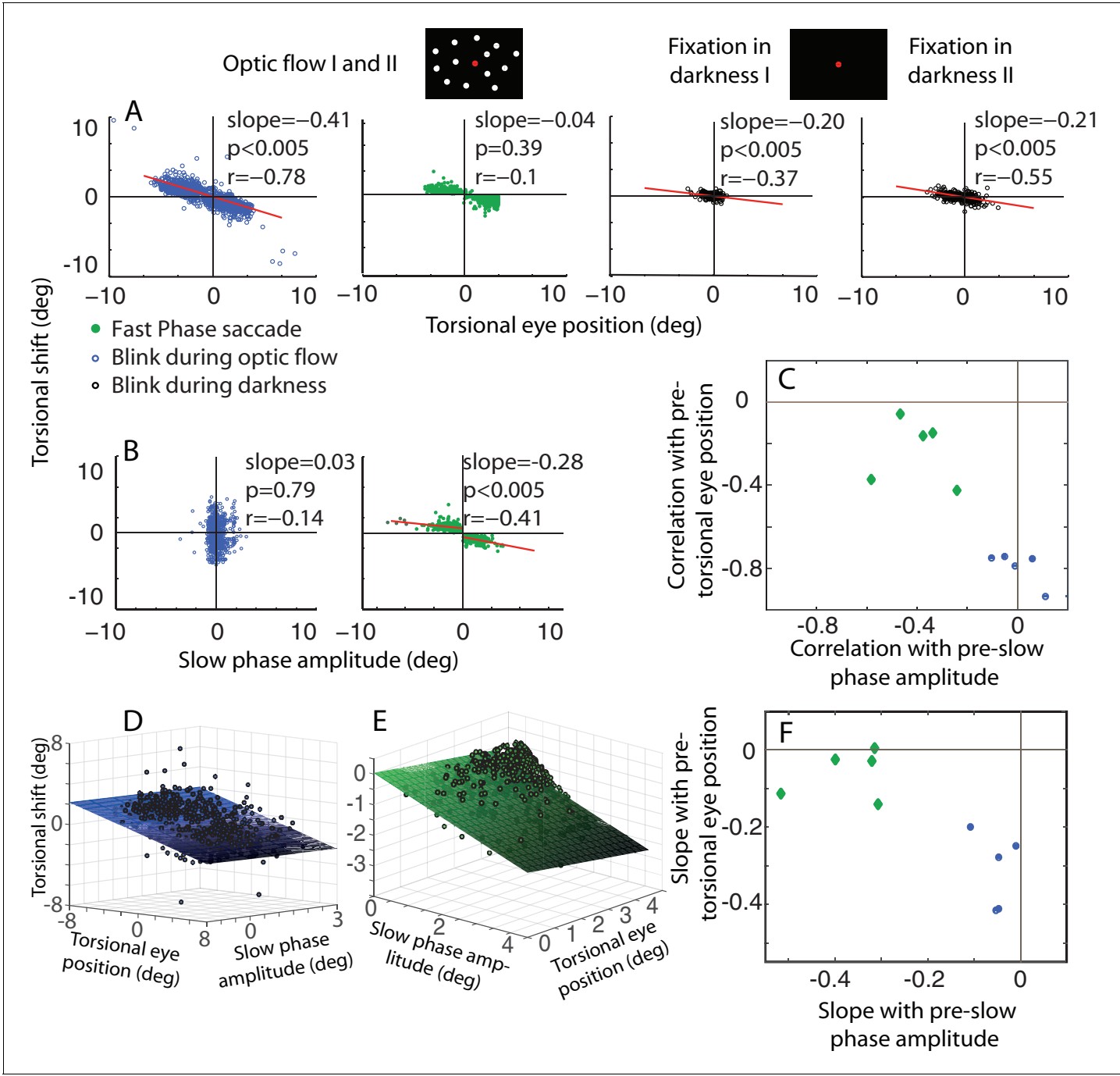

**Figure 3.** The dependence of the size of eye torsion resetting on the torsional eye position before the onset of resetting is specific for BARMs. (**A**) The torsional shift of BARMs during optic flow and darkness, and the torsional shift of fast phase saccades during optic flow is plotted as function of the torsional eye position before the shift for the subject O. Note the strong dependence of the shift size on the pre-shift torsional eye position for BARMs (blue), but not for fast phase saccades (green). This dependence of BARMs is stronger for optic flow backgrounds (blue) than for dark backgrounds (black). The regression lines (red) are only drawn in case of significant correlations (see Materials and methods). (**B**) The torsional shift of BARMs and fast phase saccades is plotted as function of the amplitude of the preceding slow phase for the subject O. In contrast to (**A**), the torsional shift of fast phases but not that of BARMs showed a dependence on slow phases. When there are two regression lines plotted the values listed refer to the mean of values derived from both regression analyses (see Materials and methods). In (**C**) the mean correlation coefficients (95% confidence interval) between torsional shift and pre-shift eye position (y-axis) are plotted relative to those between torsional shift and pre-shift slow phase amplitude (x-axis) for each of the 5 subjects separately for BARMs (blue) and fast phases (green). The clear distinction between BARMs and fast phases in their torsional shift dependence on eye position (BARMs) or preceding slow phase (fast phases) is apparent. The borders of the 95% confidence interval are not visible because they are very small (see Materials and methods). (**D**) The differential dependence of torsional shift on pre-shift eye position and pre-shift slow

*Figure 3 continued on next page*

*Figure 3 continued*
phase amplitude for BARMs is indicated by clearly different 2D linear regression fits (based on the 'optic flow' phases, subject O). The tilt of the plane and the change of the darkness of color along the torsional eye position axis illustrates the major role of pre-shift eye position for the torsional shift. (E) For fast phases (clockwise optic flow, subject O) the plane is tilted and the darkness of color changes along the slow phase amplitude axis demonstrating the major role of pre-shift slow phase amplitude for the torsional shift by fast phases. (F) Similar to (C) the slopes of the 2D regression analysis are shown for all subjects separately for BARMs and fast phases. The distinction between both is reproduced.
The following figure supplement is available for figure 3:

**Figure supplement 1.** Correlations of the torsional shift associated with blinks or fast phases with the pre-shift torsional eye position (A–C) and the preceding slow phase amplitude (D–F), respectively, for all subjects (except for subject O, shown in *Figure 3*).

are usually understood as an oculomotor behavior supporting the protection and cleaning function of blinks (*Bour et al., 2000*; *Bergamin et al., 2002*; *Straumann et al., 1996*).

Assessing the validity of these two possibilities requires information on the kinematics of the eye movements underlying BARMs. However, the eye lid closure during the blinks bars video-eye tracking from providing the information needed. This is why we repeated the experiment discussed before in five subjects, resorting to search coil recordings (*van der Geest and Frens, 2002*) (see Materials and methods). This approach allowed the precise measurement of eye position in all three dimensions independent of whether the eyes were closed or open. We observed nasal and downward eye movements during blinks very similar to previous descriptions of blink-associated eye movements during fixation in the absence of torsional optic flow (*Bour et al., 2000*; *Bergamin et al., 2002*; *Straumann et al., 1996*). The analysis of torsional eye movements during blinks showed some divergence across subjects. Four out of five subjects had long lasting blink-associated eye movements, composed of two clearly distinct components (Ha: 369 ± 54 ms; Mo 316 ± 18 ms; Da 367 ± 13 ms; Ph 372 ± 45 ms mean ± std duration). In one subject, the two components were not clearly distinguishable because they were most probably merged (see Figure 5 for reasoning).

As illustrated in *Figure 4* for subject Ha, the peak of the first component was about 130 ms from blink onset (subject Ha: 107 ± 10 ms, mean ± std, all subjects: 127 ± 35 ms) with an amplitude of about 1.2 degree (Ha: 1.6 ± 0.12 deg, all subjects: 1.14 ± 0.47 deg). The torsional dimension of the first component was always directed clockwise. Since the right eye was measured this component corresponds to an extorsional eye movement. It was supplemented by a nasal downward movement as compared to the pre-blink position. These properties match the ones previously described for blinks during fixation (*Bour et al., 2000*; *Bergamin et al., 2002*; *Straumann et al., 1996*).

**Table 1.** Relationship between torsional shifts of BARM's and pre-blink eye position for different experimental phases and ranges of pre-blink eye torsion. The steeper slopes and larger correlation coefficients for the 'optic flow' phase I and II are also observed when restricting the range of pre-blink eye torsion. Also the steeper slopes and larger correlation coefficients in the 'fixation in darkness' phase II as compared to the 'fixation in darkness' phase I are maintained when analyzing the restricted range for pooled data from all subjects and experimental blocks.

| Phase | Regression slope (mean ± std) | Correlation (mean ± std) | P-value of the correlations |
|---|---|---|---|
| fixation in darkness phase I | −0.26 ± 0.09 | −0.33 ± 0.1 | p<0.005 |
| optic flow phase I and II | −0.62 ± 0.163 | −0.73 ± 0.19 | p<0.005 |
| fixation in darkness phase II | −0.28 ± 0.15 | −0.48 ± 0.1 | p<0.005 |
| fixation in darkness phase I (−2° to 2°) | −0.08 ± 0.09 | −0.18 ± 0.14 | p<0.005 |
| optic flow phase I and II (−2° to 2°) | −0.43 ± 0.1 | −0.60 ± 0.25 | p<0.005 |
| fixation in darkness phase II (−2° to 2°) | 0.12 ± 0.07 | −0.25 ± 0.08 | p<0.005 |

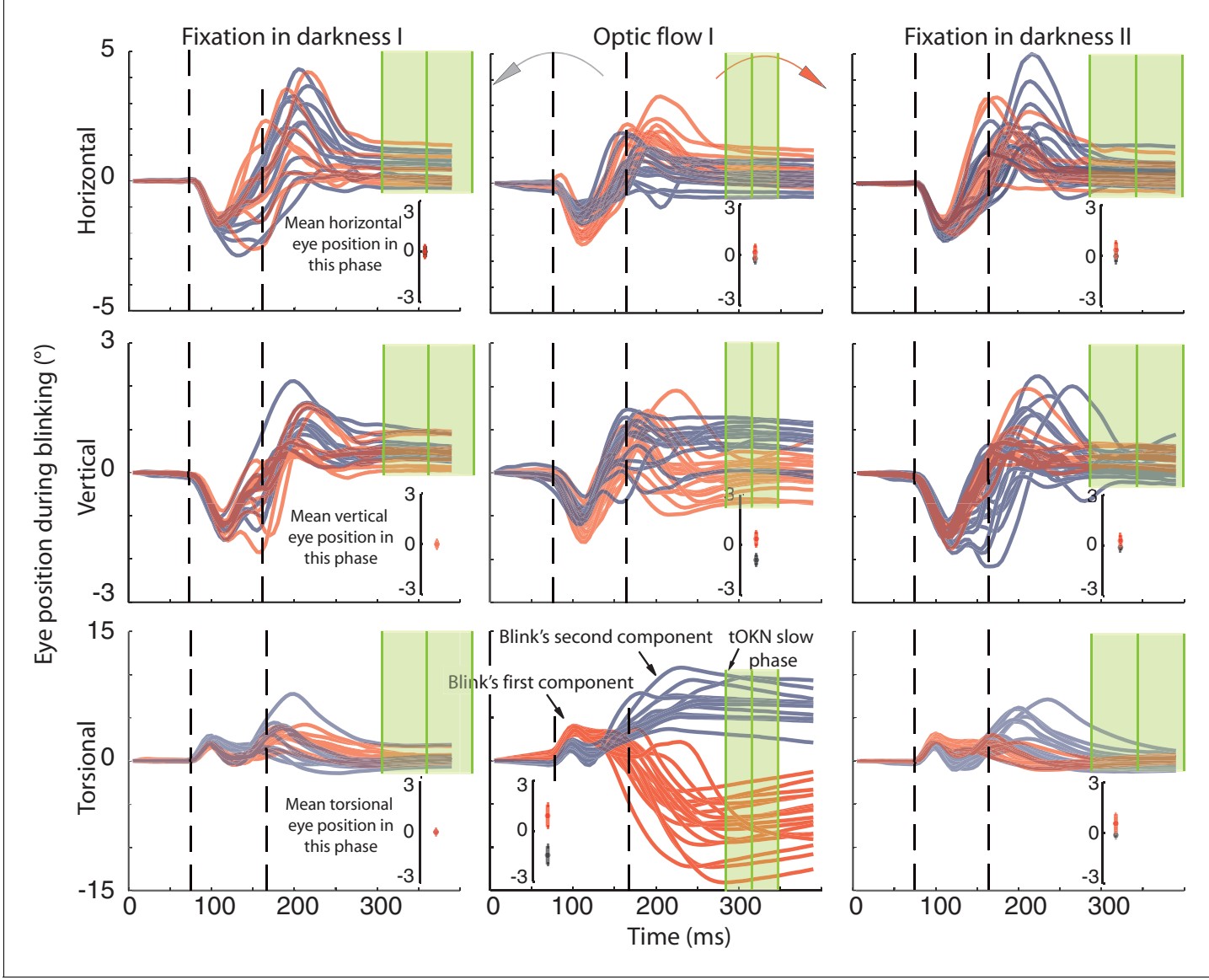

**Figure 4.** Recording of eye movements during blinks with search coils. The horizontal, vertical and torsional eye movements observed during blinks are shown in three rows. Columns represent three different experimental phases. Red traces are eye movements from blocks in which the optic flow during the 'optic flow' phase I rotated CW, grey traces indicate blocks with CCW optic flow. The same color code is used to distinguish eye movements in the two darkness phases, which preceded the 'fixation in darkness' phase I or followed by the 'fixation in darkness' 'phase II' the one with CW or CCW optic flow. To facilitate the comparison of eye movement directions during blinks with the overall eye position during a phase, the insets in each panel plot the mean eye position (± std) for the whole duration of the respective experimental phase. Note that the torsional eye movement response consists of two distinct components: the first one is positively (= extorsionally) directed and constant across the three phases, whereas the following second component is directed opposite to the overall eye position of a phase in accordance with a role in resetting the eye torsion to zero. For better visibility the torsional component is 3-times magnified (indicated by the difference in the scale of the y-axis). The eye traces are aligned to the peak velocity of the first component. Eye traces are plotted from 80 ms before and 300 ms after that time. Dashed vertical lines indicate the mean start and end of blinks obtained from the video eye tracking (see Materials and methods). To show the time points used for determining the eye position after the blink (blink end plus safety temporal distance to avoid artifacts) vertical green lines with shadows were added representing the mean ± std (across experimental blocks for Ha). Note that the second component starts during the eyelid closure and it proceeds after the point of recognizing the pupil (second dashed line), which could be during partially opened lid and not necessarily at the offset of the eyelid opening (see Materials and methods). The blink starting point taking the safety temporal distance in to account lays with around 50 ms before the onset of the figure.

Importantly, we found this first component not only in the 'fixation in darkness' phase I and II, when the subject had to fixate in darkness, but also in the 'optic flow' phase I and II when the optic flow was present. The profile of this first component did not change significantly between phases and, most noteworthy, the direction of it was not changed by the direction of the optic flow, which would be required if it had a compensatory role (see *Figure 4*). The only relation between the first component's torsional dimension and BARMs was found in 'optic flow' phase I and II, in which the torsional amplitude was larger when the optic flow pointed in the same direction, i.e. clockwise, as compared to the counterclockwise direction. This was reflected by a negative correlation between the amplitude of the first component and the amplitude of BARMs ($r = -0.43$, $p=0.0001$, based on the four subjects with two components). In other words, the tiny influence of the optic flow on the first component was congruent with the optic flow direction. For this reason it cannot account for the compensatory action of BARMs.

The opposite was found for the second component. Its peak velocity was reached $73 \pm 22$ ms (mean $\pm$ std across subjects and phases) after the velocity peak of the first component in the 'fixation in darkness' phase I and II. Although the exact velocity profile was different in the 'optic flow' phase I and II, peak velocity was reached after a similar latency (see *Figure 4*). Therefore it is justified to assume the component 2 in non-optic flow phases to be of the same origin as the one in optic flow phases. Of decisive importance is the fact that the torsional direction of the second component depended on the direction of the optic flow. As expected for a compensatory mechanism, it was always directed opposite to the optic flow. Moreover, the amplitude of the torsional component was larger during optic flow phases, when there was more need for resetting, as compared to non-optic flow phases. As a matter of fact, the torsional amplitude of the second component correlated with the amplitude of BARMs, the latter measured as in the video records by comparing pre- and post-blink eye torsion. This correlation could be seen for all phases and subjects (fixation in darkness phase I: $r = 0.63$, $p<0.05$; optic flow phase I and II: $r = 0.96$, $p<0.05$; fixation in darkness phase II: $r = 0.87$, $p<0.05$, based on pooled data from four out of five subjects). Finally, we found no consistent relationships between the amplitude of BARMs and the horizontal and vertical eye movements accompanying the blink. In sum, it is the second component, which makes the decisive contribution to the BARMs documented by video eye tracking. Moreover, the second component acts independently of accompanying vertical and horizontal eye movements. One question remains, namely if this second component of BARMs is unique to blinks or a specialized type of saccades.

From a kinematic point of view, an eye movement qualifies as a saccade if the relationship between its amplitude and peak velocity satisfies the main sequence relationship, first established for visually guided saccades [*Bahill et al., (1975)*]. A plot of the fast phases of the tOKN in the 'optic flow' phases I and II showed the expected monotonic, almost linear growth of peak velocity with amplitude (*Figure 5*). The two blink components differed in their location on the velocity-amplitude plot and in the relationship between amplitude and peak velocity. While the first component of blink-associated eye movements recorded with search coils did - as expected - not show a consistent dependency of peak velocity on amplitude for all subjects, the second component (=BARM) exhibited the typical relationship of a saccadic eye movement not that different from that of fast phases although the regression line was significantly shifted to slower velocities ($p<0.05$ for all subjects) (see Materials and methods, and *Figure 5*). Hence, in kinematic terms BARMs are a special type of saccade that can be distinguished from fast phases because of their very different main sequence profile.

## It is not the blink-induced loss of visual drive that resets the eyes during blinks

One possible reason for the occurrence of BARMs during optokinetic stimulation could be that the blink inevitably interrupts optic flow, the driving force for the slow phase of the tOKN. To test this 'afference hypothesis' we simulated the altered visual input during blinks by temporarily blanking the optic flow pattern during the 'optic flow' phase I and II for durations corresponding to the ones of typical blinks (see *Figure 6* and *Figure 6—figure supplement 1*). Although we obtained significant correlations between the size of the torsional eye shift during blanking and the eye torsion before blanking in 4 out of 5 subjects, the slopes of the linear regressions were almost zero in all of five subjects. Thus, the lack of optic flow input during blinks cannot explain the occurrence of BARMs. In line with this conclusion, we did not find any correlation between blink duration and the

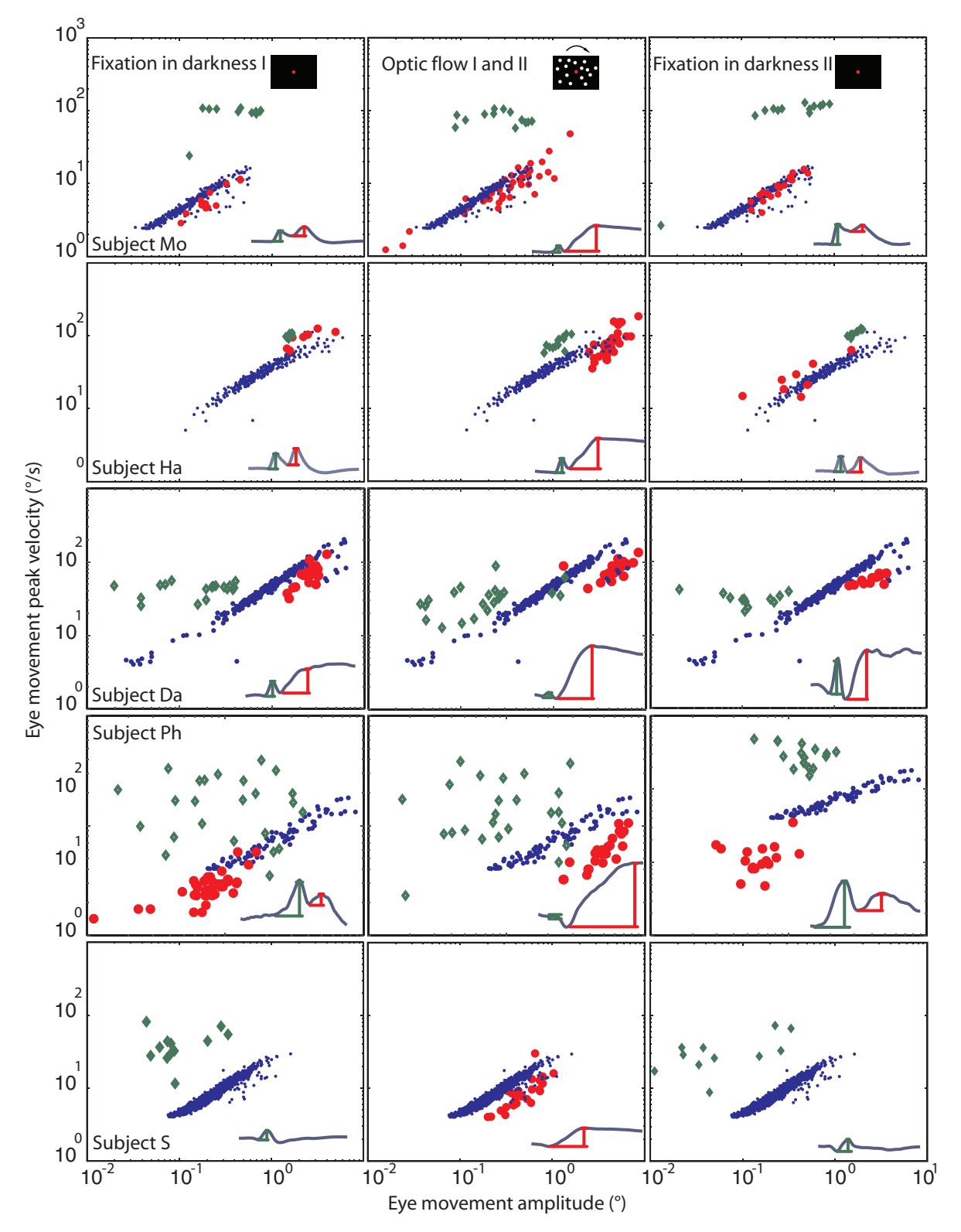

**Figure 5.** Eye movement peak velocity as a function of BARMs amplitude. The peak velocity of the eye movements is drawn on the y-axis vs. their amplitude on the x-axis. The first component of blink related torsional movement is drawn in green diamonds, the second component (BARMs) in red circles, and the fast phase of tOKN in blue dots. The 5 rows show search coil recordings from 5 subjects. The 3 columns represent the consecutive experimental phases with the exception of the fast phases (blue) which were recorded in the 'optic flow' phase I and II but are additionally plotted in

*Figure 5 continued on next page*

*Figure 5 continued*

the other two phases to facilitate the comparison. In subject S, wherein the two components of the eye movement are hardly distinguishable the eye movement is considered as first component in the darkness phases I and II but as second component in the optic flow phases. Accepting this qualification the two components of all subjects form two separate distributions. Both are different from the main sequence of fast phases with the first component lacking a consistent velocity-amplitude relationship and the second component having a smaller slope of the velocity-amplitude relationship than fast phases.

amplitude of BARMs, neither for true nor for simulated blinks, a relationship one might have expected if it was the lack of optic flow that drove the eye back to zero torsion (see *Figure 6* and *Figure 6—figure supplement 1*).

## Generalization to horizontal and vertical eye movement components

We finally asked if BARMs are confined to eye torsion. Alternatively, also deviations from the horizontal or vertical zero position might be corrected by appropriate blink-associated eye movements. When considering each experimental phase and subject separately, measurements by video-oculography did not exhibit any consistent correlation between blink-associated eye position shifts and pre-blink eye positions, neither for the horizontal nor for the vertical dimension.

We considered the possibility that the lack of significant correlations in individual subjects and phases might have been a consequence of the limited spatial resolution of the video-eyetracker, due to the impact of small residual head movements made by subjects. Therefore, we also analyzed eye data obtained with the search coil technique, offering a much higher spatial resolution (see Materials and methods for details). As a result, we now consistently found a significant correlation between eye position shifts and pre-blink eye positions for both, horizontal and vertical dimensions as well as for the torsional dimension (see *Figure 7*).

The relationship between pre-blink eye position and the amplitude of vertical BARMs is particularly apparent in subject HA (see *Figure 4*, central). He had a mean vertical deviation of $-0.87 \pm 22$ deg during CCW and $0.42 \pm 0.36$ deg (mean $\pm$ std) during CW optic flow stimulation. Each of his blinks ended in the opposite direction of the mean vertical deviation before blink, leading to a clearly visible difference of vertical eye positions after blinks dependent on the optic flow direction. This is remarkable in view of the small vertical deviations of less than 1 deg. BARMs are obviously sensitive enough to compensate for them.

## Discussion

In this study we have demonstrated the existence of a distinct, hitherto unknown type of eye movement. It helps to avoid the accumulation of excessive eye torsion, building up in response to torsional optic flow by time and again resetting the torsional deviation. While this resetting eye movement serves a similar purpose as the well-known torsional fast phase component of the torsional optokinetic nystagmus (tOKN), it is different from the latter for several reasons: 1. This new eye movement is a key element of a motor synergy in which it is yoked with eye blinks, which is why we refer to it as blink-associated resetting eye movement (BARM). 2. The horizontal and vertical movement components complementing the torsional resetting response are qualitatively different from those of the fast phases of the tOKN. 3. The amplitude of BARMs is correlated with the pre-blink torsional eye position but not with the amplitude of the preceding slow phase component of the tOKN, a relationship, which is exactly opposite to the fast phases of the OKN as also reported before (*Watanabe et al., 1994*; *Waddington and Harris, 2012*). 4. BARMs differ in the main sequence relationship from tOKN fast phases. 5. BARMs are also observed in the absence of an optokinetic stimulus in case significant amounts of eye torsion may have arisen spontaneously. 6. Similar blink-associated resetting eye movements are deployed to reset eye deviations from straight ahead in the vertical and horizontal dimension. In sum, we suggest that BARMs are a distinct type of eye movement that is a ubiquitous partner of eye blinks.

As said before, BARMs were found independent of the specific condition that had led to eye torsion, consistently across subjects and independent of the two eye movement recording methods we used, video eye tracking and search coil recordings. Video eye tracking has the advantage of

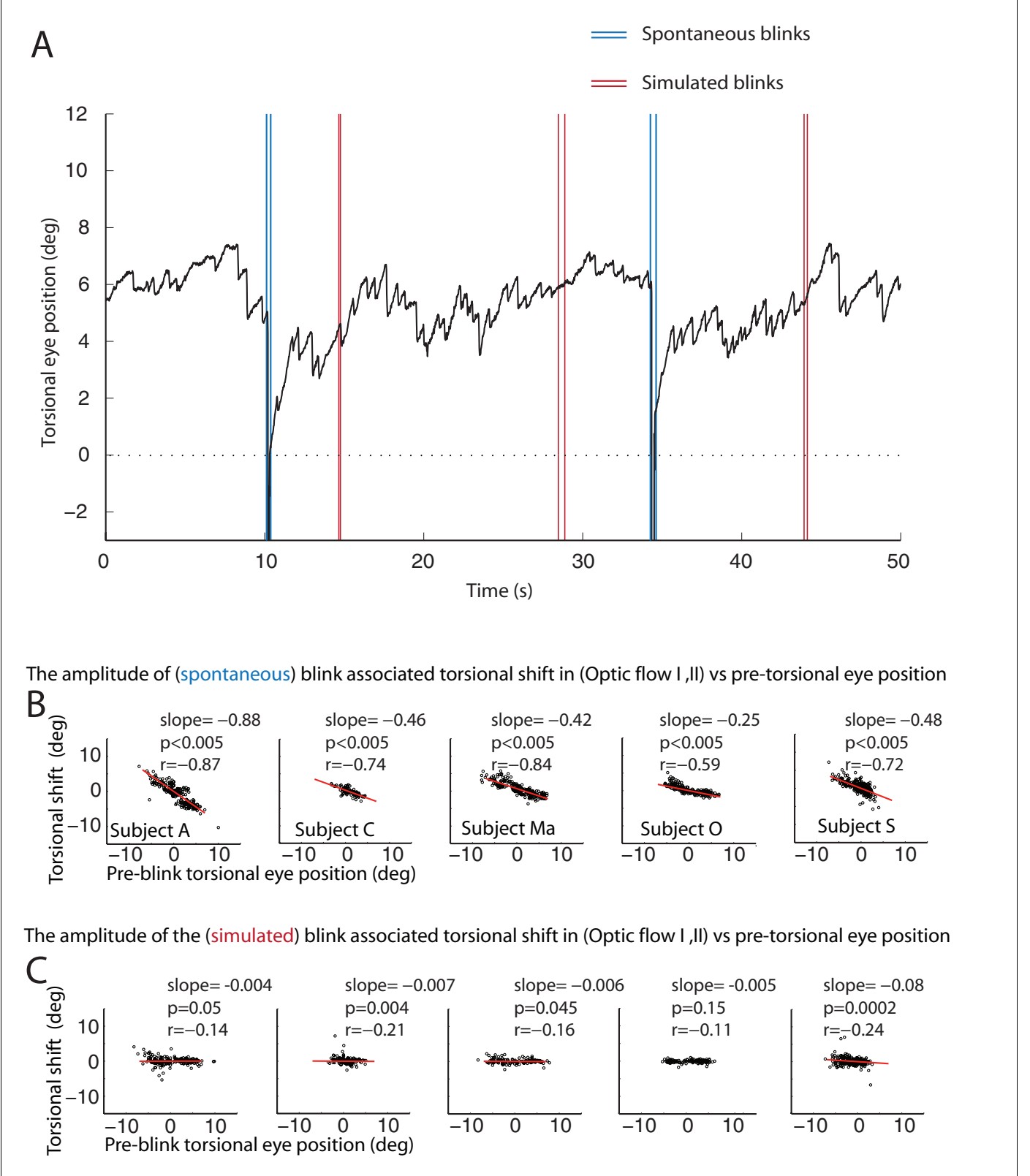

**Figure 6.** The resetting of the eye torsion is associated with spontanous blinks and not due to stopping the optic flow. (**A**) Torsional eye position before and after spontaneous and simulated blinks. Onsets and offsets of spontaneous blinks (blue) and simulated blinks (red). Spontaneous as well as simulated blinks had variable durations. Note change of the torsional eye position only after spontaneous but not after simulated blinks. (**B**) The torsional shift associated with spontaneous blinks drawn against the pre-blink torsional eye position. Note the significant correlations. (**C**) The torsional

*Figure 6 continued on next page*

*Figure 6 continued*

shift associated with simulated blinks drawn against pre-blink torsional eye position. Note that although the correlation between the torisonal shift and the pre-blink torsional eye position for the simulated blinks is significant in some subjects the slope is almost zero.

The following figure supplement is available for figure 6:

**Figure supplement 1.** Correlations of torsional eye position shift and the duration of spontaneous and simulated blinks during torsional optic flow.

avoiding any interaction with the eye and the eye lids. On the other hand, it does not provide information for the time the eyes are covered by the eye lids. Complementary search coil recordings allowed us to fill this gap and to document a sequence of two movement components. Whereas the first one does not account for the properties of BARMs, the second one does. Perhaps one might object that the second component is only an artifact of the search coil recording technique, e.g. due to the need to anesthetize the eye (*Bergamin et al., 2002*, *2004*). Yet, the congruency of the overall amount of resetting documented with video eye tracking and search coil recordings clearly speaks against this concern.

There are two additional aspects of BARM control which need to be considered. Firstly, although BARMs reset eye torsion, this resetting is not complete if torsion is induced by torsional optic flow. This stands in contrast to torsional deviations observed in the absence of torsional optic flow where we observed more BARMs leading the eyes back to zero. A possible explanation of the incomplete resetting of optic flow induced torsion might be the lack of naturalness of the condition. Continuous torsional optic flow in one direction, present for 270 s as in our experiments, might result from being spun in a leisure park attraction but hardly from ego motion in a natural environment. Actually it is tempting to speculate that the building up of a torsion plateau in reaction to the exposure to steady state torsional optic flow might reflect an adaptive shift of the subjective vertical. Interpreted this way, BARMs would not be driven by the deviation of the eyes from a baseline zero but by the deviation from the adapted torsional zero (*Thilo and Gresty, 2002*; *Ibbotson et al., 2005*).

The second aspect is that the influence of the level of pre-blink eye torsion on the amplitude of BARMs depends on context and history. This is indicated by slight, yet significant differences of the BARM-amplitude vs. pre-blink eye torsion regressions. The influence was strongest - as expressed by the steepest regression line slopes as well as the largest coefficients of correlation - for experimental phases which led to higher initial levels of torsion due to a build-up of torsion as in the 'optic flow' phase I and II. This modulation of the slope is actually quite useful as it helps to counteract the accumulation of eye torsion in the presence of strong torsional optic flow.

Are BARMs simply inevitable passive biomechanical consequences of blinks? While blinks may have an impact on eyes that deviate from straight ahead, the rotation symmetry of the eyes suggests a mechanical reaction that should be independent of eye torsion. Yet, BARMs are always compensatory, which requires that they consider information on the initial torsional position in order to select an appropriate amplitude and velocity as well as a relationship between the two that is reminiscent of a saccade-like main sequence characteristic. Hence, the evidence available clearly argues for active control of BARMs rather than a biomechanical artifact. It is an active process linked to blinks rather than being elicited by an interruption of vision as indicated by control experiments in which we simulated the blink-associated interruption of the visual input.

The second blink-associated eye movement component seen in search coil recordings which is largely rotational, lacking significant accompanying horizontal or vertical eye movements corresponds to BARMs as seen in video eye tracker recordings. The first component corresponds to the classical blink-associated eye movement which is directed nasally, downward and extorsional - a reflex that has been suggested to support the hydration and protection of the eye (*Evinger et al., 1991*; *Evinger, 1995*). The independence of the two components is documented by the consistent direction of the first independent of the resetting direction of torsion, and the absence of the consistent amplitude vs. velocity relationship that characterizes the second component, an independence that argues that the latter is a rigid blink-specific eye movement that is not related to other types of oculomotor behavior serving vision. Although previous work on search coil recordings of blink-associated eye movements had noted the occurrence of a second component, it did not address its function (*Bergamin et al., 2002*).

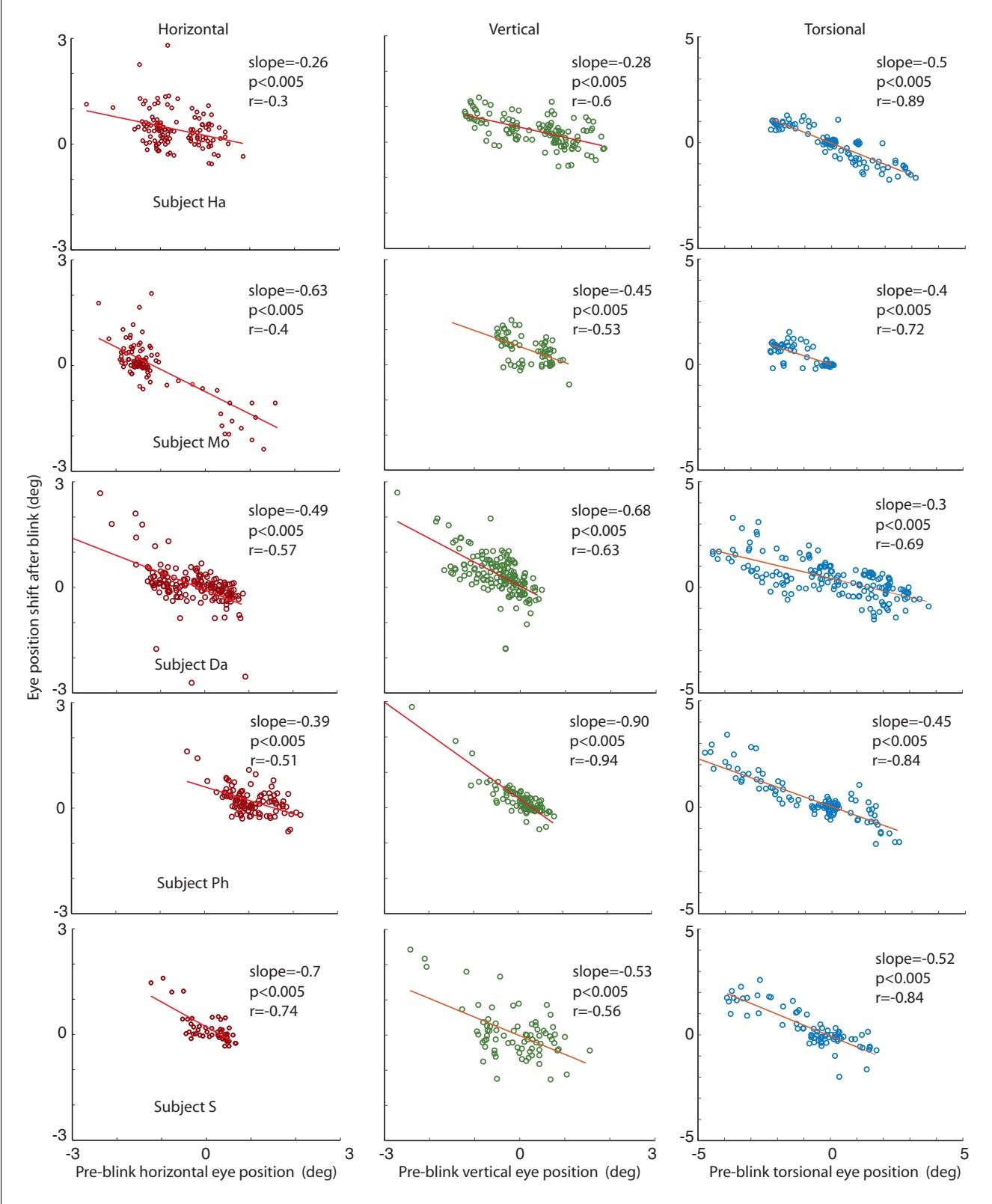

**Figure 7.** Correlations of the amplitude of the horizontal, vertical and torsional shifts after blinks with pre-blink eye position recorded using the eye coil. The horizontal eye position shift after blinking is plotted against the horizontal pre-blink eye position for 5 subjects in the first column (red circles). The vertical eye positions are plotted in the second column (green circles), and the torsional in the third (blue circles). Note that there is a significant correlation of the size of the shift after the blink with the pre-blink eye position.

Is the second component underlying the BARM a slowed down saccade, spread out over the whole period of the visual gap resulting from the eye blink? Reducing saccadic speed as much as possible would satisfy demands of energy efficiency while not further burden vision. Changes in saccade velocity have actually been observed in response to a wide variety of conditions (*Prsa et al., 2010*; *Shadmehr, 2010*). Most importantly, it is well-known that saccades which coincide with blinks have latencies and kinematics that differ from those of non-blink-associated saccades (*Goossens and Van Opstal, 2000*; *Powers et al., 2013*; *Rottach et al., 1998*; *Rambold et al., 2002*).

On the other hand reports on patients suffering from pathology of the brainstem premotor circuits for saccades may actually suggest a saccade-independent nature of BARMs. Unlike healthy subjects, these patients exhibit slow saccades which may be faster if they are associated with a blink. Actually in extreme cases only blink-associated saccades may be left, while others are lacking completely (*Zee et al., 1983*). Rather than assuming a facilitatory influence of blink-related signals on a run-down saccade generator, a more parsimonious interpretation could be that the putative blink-associated saccades in these clinical case are actually BARMs, still possible because the underlying circuitry may have been spared by the pathology. On the other hand, the blink-associated oscillations that have been observed in patients suffering from neurodegenerative brainstem disease with normal saccades (*Hain et al., 1986*) might be a manifestation of a degenerated BARM generator sparing the saccade generator. In other words, the speculative conclusion would be the assumption of a far-reaching dissociation of the generators of BARMs and saccades. The practical implication of this scenario could be that patients suffering from a loss of saccades might actually train to deploy BARMs more frequently and efficiently.

In summary, blink-associated resetting eye movements represent a distinct type of eye movement intimately linked to eye blinks, helping the observer to keep the deviation from a baseline torsional eye position small, thereby supporting stable spatial vision. While we explored the properties of these eye movements primarily in the torsional domain, preliminary evidence suggests that they may not be restricted to torsion but may help to keep also horizontal and vertical deviations at bay. Yoking these eye movements with the closure of the eyelids is an ecologically useful synergy that helps to minimize the downtime of vision that is inevitably associated with both. Considering that we blink on average 18 times per minute over 16 hr of wakefulness per day and furthermore that each BARM makes a resetting saccade of 50 ms length waivable, the gain in uncompromised vision is substantial, amounting to 15 mins every day.

## Materials and methods

### Subjects

Eleven human subjects without known neurological diseases participated in the experiments of this study (3 females, 8 males, age 25–33). One participated in experiments 1, 2, and 3 (subject S), two of them in two experiments (subjects: A and O) and the others in just one out of the three experiments (subjects: M, D, C, Ma, Mo, Ha, Da, and Ph). All subjects had normal or corrected-to-normal visual acuity and gave written informed consent according to the declaration of Helsinki prior to the experiment. The study was approved by the institutional ethics committee.

### Experimental setup and stimulus

All experiments were carried out in darkness. Subjects sat with their head gently restrained by a chin and forehead rest for the duration of the experiments at a distance of 60 cm from the stimulus display. The stimuli were presented on a large frontoparallel tangent screen (160 cm width and 120 cm height, $108 \times 90$ deg visual angle). A NEC GT2150 projector with a resolution of $1280 \times 1024$ pixels, a frame rate of 60 Hz, a luminance of 2500 lumen, a response latency of $60 \pm 0.7$ ms (mean ± std measured in our lab using a photodiode), and a contrast ratio of 1:350 was used to project the stimulus on the screen. The center of the screen was aligned to the naso-occipital axis of the subject. A central red fixation dot (0.3 deg diameter) was located there. Each experimental block consisted of sequences of phases of in which no stimulus was available other than the fixation point and phases in which torsional optic flow (tOKN) was present. In any case, subjects were encouraged to fixate the red fixation dot available during both phases. The tOKN stimulus consisted of a random dot pattern

(1500 dots, 0.3 deg diameter of each, density 0.55 dot/deg$^2$) rotating around the naso-occipital axis at an angular speed of 30 deg/s, either clockwise or counterclockwise.

## Experimental design

Prior to each session the eye position output provided by the video eye tracker (Chronos Eye Tracking System, Chronos Vision GmbH, Berlin, Germany) in all experiments and that of an additional 3D search coil system deployed in experiment 3 were calibrated in 2D by using a routine, requiring the subject to maintain fixation of targets dots presented at random in nine locations on a 3 × 3 grid with a spacing of 5 deg. Each dot, visible for 1 s, had to be fixated at least 3 times each. The torsional component of the search coil signal was calibrated offline with the eye torsion measurements provided by the video eye tracker operating in parallel with the search coil recordings.

### Experiment 1

Each block in this experiment consisted of four phases. The first one required fixation of the central target in complete darkness for 180 s. After that a second phase started in which the torsional optokinetic stimulus was presented, rotating in experiments on a given day consistently either in clockwise or counterclockwise direction for 270 s. The third and fourth phases were repetitions of the first and the second ones respectively. The subjects performed 16 blocks in total with 2 blocks a day. The direction of optic flow for a given subject and day was chosen pseudorandomly with the qualification that in the end both directions needed to be represented equally in the data pool.

### Experiment 2

This experiment was identical to experiment 1 except that the stimulus phases included a simulation of the loss of visual drive associated with blinks. This was achieved by blanking the optic flow pattern for different durations (100, 200, 300, 400, and 500 ms) spanning the range of observed blink durations (*Figure 6* and *Figure 6—figure supplement 1*). The residual luminance during blanking the stimulus was about 0.3 cd/m$^2$. Although less light might reach the retina during a real blink, the critical variable is not luminance but optic flow. And optic flow was completely eliminated by blanking. The blinks occurred based on probabilistic functions having a half sine wave profile peaking at 5 s with width of 5 s leading to inter blanking intervals between 0 to 10 s. The blanking had different durations reaching from 100 to 500 ms.

### Experiment 3

This experiment corresponded to two blocks of experiment 1 with the qualification that between the two blocks, the direction of flow was switched from clockwise to counterclockwise. The major difference was that the eye movements were measured with 3D search coils in parallel with video-oculography, the latter being the only eye movement recording method used in the other two experiments.

## Eye movement recordings

### Video-oculography

We recorded eye movements deploying an infrared video eye tracking system (Chronos Eye Tracking System, Chronos Vision GmbH, Berlin, Germany) with 50 Hz or 100 Hz. As the major interest in experiments 1 and 2 was the documentation of slow changes in torsional eye position and the consequences of resetting the torsional eye position, a sampling rate of 50 frames/s turned out to be sufficient. This sampling rate was also fast enough to allow us to reliably detect blinks.

The horizontal and vertical eye positions of the left and right eyes were directly derived from the eye tracking system, which basically detected the position of the pupil by a contrast threshold on the 2D luminance histogram of the eye pictures. The size of the pupil was 62 ± 16.4 pixels (mean ± std, calculated based on information provided by the video eye tracking software and then averaged across subjects and phases). The torsional eye position was computed offline using the iris tracking software of the system (Chronos Vision GmbH) which determines the rotation relative to a reference frame based on a cross-correlation algorithm (*Ong and Haslwanter 2010*).

Eye blinks were detected by the fact that the transient coverage of the pupil led to the omission of several subsequent samples, recognized by accompanying infinite values. Detected events were

considered true blinks only if they appeared in both eyes simultaneously for all eye positions (vertical, horizontal and torsional) thereby distinguished from occasional pupil detection artifacts. The start of a blink was defined by the appearance of infinite values resulting from a failure of pupil recognition due to the lid closure. Correspondingly, the end of a blink was defined by the reappearance of finite values. The means of start and end of blinks for subject Ha are illustrated by the dashed lines in *Figure 4*. The reappearance of finite values is a consequence of the reappearance of the pupil. However, the resulting eye position data for the moments of transition from lid open to closed and back again are usually unreliable as the detection algorithm is easily fooled by still partially covered pupils. We therefore introduced a temporal safety distance (7 frames before and 10 frames after) relative to these moments of instability and measured the eye position 140 ms before and 200 ms after the blink. Note that our method to detect the blink's onset and offset is approximate as it depends on the pupil recognition and not the diction of the earliest lid movements.

We checked if changing the interval length had an influence on the data, which was not the case.

### 3D search eye coil system

3D scleral search coils from three different vendors were used: Chronos Vision, Berlin, Germany, Skalar from Delft, Netherlands and Universal trading ventures, Cleveland, USA, were used in the third experiment measuring the high speed movements made during blinks. The data were sampled at 200 Hz.

Stimulus generation, data acquisition and experiment control was realized using the open source measurement system nrec (http://nrec.neurologie.uni-tuebingen.de, created by F. Bunjes, J. Gukelberger et. al.) running on a standard debian Linux PC.

## Statistical analysis

All data were analyzed using Matlab 2012 aroutines (MathWorks, Natick, MA).

### Correlation and regression analyses

Correlation coefficients (Pearson) and regressions were calculated between the size of eye torsion resetting (torsional shift) and torsional eye position before the onset of resetting for both, shifts associated with blinks (BARMs) and shifts due to fast phases of tOKN. In the same way the relationship between torsional shifts and the amplitude of the preceding torsional slow phase was calculated. For BARMs the slow phase amplitude was determined as change in torsional eye position from the end of the preceding fast phase to the onset of the blink. Since torsional shifts can have two directions (clockwise or counterclockwise), depending on the direction of optic flow direction, a correlation or regression analysis that relies on samples pooled over both directions may erroneously suggest a linear relationship simply if the data points related to the two distributions form sufficiently distinct clusters with data points within clusters positioned at random. To test whether the data points are continuum, forming one cluster, or if they are separated, forming two clusters depending of the optic flow direction, we did the following: first, the analyses were primarily done separately for each direction of torsional shift. Then we repeated the same analysis but for having the data as one pool. Finally, We compared the slopes and offsets from the regression analysis applied to pooled data for both directions with the ones for the analysis of the two pools sorted by direction. For the torsional shifts associated with fast phases we found large differences of offset and slopes for the one cluster vs. two cluster analysis. On the other hand, the shifts of BARMs had almost equal slopes and offsets for both types of analyses. Take for example subject O (shown in *Figure 3A and B*), the slope difference for shifts by fast phases was 0.18 and the offset difference was 1.4°. In contrast, for blinks the slope difference was 0.021 and the offset difference 0.14°. The same analysis was applied to pooled data from all subjects. The regression slope difference was $-0.21 \pm 0.1$ (mean ± std) for fast phases and $0.025 \pm 0.039$ for blinking. The offset difference was $1.6 \pm 0.7°$ (mean ± std) for fast phases and $0.19 \pm 0.35°$ for blinking. The differences were smaller for blinks by a factor of 10. In view of these results, we pooled the data from both directions for analysis of BARMs, but analyzed fast phases separately for each direction. Single values used or given for fast phases (for example in *Figure 3A*) represent the mean of values for both directions.

Similarly, the relationship between the torsional shift by fast phases and the pre-shift slow phase amplitude was influenced by the direction of the optic flow. For example the regression slope

difference was around 0.4 for subject O with an offset difference of 1.13 shown in *Figure 3B*. For all subjects the regression slope difference was on average -0.4190 ± 0.0826 (mean ± std) and the offset difference of 1.5802 +- 0.9375° (mean ± std) (see *Figure 3—figure supplement 1F*). Similar to A, single values used or given for fast phases (for example in *Figure 3B*) represent the mean of values for both directions

For the statistical analysis underlying *Figure 3C*, we pooled the torsional shifts by fast phase tOKN, during BARMs, pre-blink and pre-fast phase torsional eye position, and the amplitude of the preceding slow phase from all blocks and all phases of every subject studied in experiment 1. For each pool of data, contributed by individual subjects, we had more than 400 samples. We generated bootstrapping pools per subject by taking 400 samples with replacement randomly from each pool for 1000 times and correlating them as follows: 1. The torsional shift of fast phase tOKN with their related preceding slow phase amplitude and pre-fast phase eye position, 2. The torsional shift of BARMs with their related preceding slow phase amplitude and pre-blink torsional eye position. These are four different types of correlations with two dependent variables, torsional shifts from fast phase tOKN and from BARMs. The relationship of both dependent variables (differing in color) on pre-shift eye position on the one hand and pre-shift slow phase amplitude on the other is represented by the mean correlation coefficients and their 95% confidence intervals across 1000 correlations per subject in *Figure 3C*. In *Figure 3F*, instead of the correlation coefficients, we plot the means of the slopes of fitting the data with 2D linear regressions (torsional shift = a · pre-shift slow phase amplitude + b · pre-shift eye position) and their 95% confidence intervals are shown for the same relations. For the reasons discussed above, the torsional shift associated with BARMS were fitted with one model for both directions whereas torsional shifts from fast phases were fitted with two separate models.

## Additional analysis: Blink-associated resetting of eye torsion is realized by a specific eye movement

As mentioned in the results only two subjects had long blink durations with a clear distinction of two components. In the third subject (S), whose blink duration was 288 ± 45 ms (mean ± std), the two components could not be distinguished reliably. However, as shown in *Figure 5*, the timing of the torsional peak depended on the experimental phase. During fixation in darkness phase I and II, in which the need for a resetting eye movement is low, the torsional peak was seen earlier than in the 'optic flow' phase I and II (see *Figure 5*). Assuming that both components are merged in this subject it suggests that the first component dominated during fixation in darkness, when there was almost no need for compensation, and consequently, less need for the second component. In contrast, during the 'optic flow' phase I and II, when the torsional peak was later, its velocity-amplitude relationship was shifted to lower velocities (18.6 ± 4.5 mean ± std, fast phases: 31.5 ± 5.1 deg/s) compared to fast phases resembling the second component as seen in the other four subjects (BARMs in Mo: 32.6 ± 10.5, in HA: 19.5 ± 5.44, in Da: 17.5 ± 8.3, in Ph: 13.3 ± 4.27; fast phases in Mo: 38 ± 9.16, in HA: 23.5 ± 5.6, in Da: 42 ± 15.2, in Ph:; 23.2 ± 7.5 significant difference between BARMs and fast phases for each subject, Wilcoxon rank-sum test p<0.05).

## Additional analysis: Generalization to horizontal and vertical eye movement components

We compared the noise level of our video eye tracking system and our homemade search coil system. The noise level was assessed by calculating standard deviations of the eye position during the whole phase of 'fixation in darkness' phase I in 3 dimensions per block. To obtain a value for each subject, the mean across blocks was calculated per subject. For the video eye tracking system the amount of noise was comparatively high in the horizontal and vertical dimensions (4.8 ± 1.3° mean ± std across subjects, 6.4 ± 3.8°, and 1.6 ± 0.34° for horizontal, vertical, and torsional, respectively). A major reason for the high level of noise seen in the video eye tracking system for the horizontal and vertical eye position are inevitable tiny translational head movements, which cause a translation of the pupil image relative to the camera that is misinterpreted by the algorithm as eye movement. Torsional eye position is less noisy because torsional head movements are smaller. The amount of noise was in general much lower for the coil system (0.29 ± 0.15°, 0.29 ± 0.12°, and 0.115 ± 0.075° for horizontal, vertical and torsional eye positions). The fact that in particular the translational noise

components are smaller is a consequence of the fact that the signal is invariant to small changes in head position along the translation axes.

## Acknowledgements

Thanks to the members of the PT lab for their help. We are grateful for the straightforward support with search coils provided by Dominik Straumann, Zürich. This work was supported by the DFG Research Unit FOR 1847, project A3 (TH 425/13-1).

## Additional information

### Funding

| Funder | Grant reference number | Author |
| --- | --- | --- |
| Deutsche Forschungsgemeinschaft | Research Unit FOR 1847, project A3 (TH 425/13-1) | Peter Thier |

The funders had no role in study design, data collection and interpretation, or the decision to submit the work for publication.

### Author contributions

MFK, Conception and design, Acquisition of data, Analysis and interpretation of data, Drafting or revising the article; JKP, Analysis and interpretation of data, Drafting or revising the article; AS, Collected around 30% of the data and processed most of the video based eye tracking data in order to extract the torsional eye position component - this task involved identifying a spot on the iris that would allow stable tracking, Contributed to writing the paper by preparing several of the figures and taking over a significant part of the Methods and materials section; FB, Acquisition of data, Analysis and interpretation of data, Drafting or revising the article; PT, Conception and design, Analysis and interpretation of data, Drafting or revising the article

### Ethics

Human subjects: All subjects gave written informed consent and consent to publication according to the declaration of Helsinki prior to the experiment. The study was approved by the ethics committee of the University of Tuebingen.

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
