## [Decision Letter]

Thank you for submitting your article "A new motor synergy that serves the needs of oculomotor and eye lid systems while keeping the downtime of vision minimal" for consideration by *eLife*. Your article has been reviewed by three peer reviewers, one of whom, Jennifer Raymond, is a member of our Board of Reviewing Editors, and another is Jacqueline Gottlieb (Reviewer #2). The evaluation has been overseen by David Van Essen as the Senior Editor.

The reviewers have discussed the reviews with one another and the Reviewing Editor has drafted this decision to help you prepare a revised submission.

Summary:

This paper reports the discovery of a new type of eye movement, referred to as blink-associated resetting movement (BARM). BARMs are rapid eye movements that correct deviations in eye position from neutral position. BARMs occur during optokinetic nystagmus and, to a lesser extent, during fixation of a stationary target. Because they are coordinated with eye blinking, BARMs correct eye position drift in a way that minimizes interference with vision. The properties of BARMs are distinct from saccades or the fast phases of optokinetic nystagmus, and thus appear to be a truly novel type of eye movement. Most of the analysis of BARMs was done for torsional eye movements, but BARMs also seem to occur in the horizontal and vertical eye movements.

The reviewers agreed that the experiments are sound and carefully done. The results represent a significant advance in our understanding of motor coordination.

Essential revisions:

1) The kinematic measurements obtained with eye coil recordings provide important support for the argument that BARMs represent a truly novel type of eye movement. More analysis and possibly eye coil data from more subjects are needed to strengthen this portion of the manuscript. Figure 5 reveals considerable variability across the three subjects recorded with eye coils, and much of the text describing the two kinetically distinct components of the blink-related eye movements seems to be describing results from the one subject shown in Figure 4 rather than a summary of results from the three subjects. Only one subject (S) seems to have been tested with both eye coil and video-oculography, and this subject shows atypical blink-related eye movement, making it difficult to compare the results obtained with the two methods. It is important to more clearly document which of the observations made with eye coils were consistent across subjects.

2) There are several additional places where the statistics need to be clarified:

a) Although the statistics reveal no significant correlation between the torsional eye position just before the fast phase of optokinetic nystagmus and the amplitude of the quick phase (green in Figure 3 and Figure 3—figure supplement 1), inspection of the plots suggests that for a subset of the quick phases, there is a correlation with the eye position. The authors should comment on this.

b) Figure 3, the authors should clarify whether this is population or individual subject data, and why the data on fast phase saccades are fit with two lines, but there is a single correlation coefficient listed.

c) Figure 3 shows the correlation coefficients with significance intervals estimated for each subject using a bootstrap analysis. A bootstrap analysis gives the standard error of the mean, and the customary use is to show the 95% confidence interval (the interval that includes 95% of the bootstrapped coefficient). Also, was the bootstrap analysis done with or without replacement?

d) Table 1 requires a description of the numbers given (what do the p-values compare? Do the numbers show means, medians, standard deviation, standard error, etc.).

e) The authors show separate correlations between BARM amplitude and (1) pre-blink eye position, (2) pre-blink slow phase excursion. Wouldn't it be better to fit the BARM amplitude to a regression containing several terms (related to position, excursion and their interaction) and show the coefficients related to each term?

f) The data for subjects A and S seem to be repeated in Figure 3 and Figure 6, but the numbers are slightly different in the two figures. The authors should comment on why this is the case.

[Editors' note: further revisions were requested prior to acceptance, as described below.]

Thank you for resubmitting your work entitled "A new motor synergy that serves the needs of oculomotor and eye lid systems while keeping the downtime of vision minimal" for further consideration at *eLife*. Your revised article has been favorably evaluated by David Van Essen (Senior Editor) and a Reviewing Editor.

The manuscript has been improved but there are some remaining issues that need to be addressed before acceptance, as outlined below:

1) The group data for the result shown by the single example in Figure 6, which are currently shown in Figure 6—figure supplement 1 and C should be included in the main figure rather than a supplementary figure.

2) Please double check the values for the slopes in Figure 3—figure supplement 1. The numbers given for the slopes don't seem to correspond well with the slopes of the red regression lines, although this could be the result of fitting not one, but two regression lines to the data, and giving the mean of the two slopes. Even considering the latter, the relative slopes for the first and second panel from the left do not seem to correspond to my visual estimation of the relative slopes from the red lines – it may just be my eyes, but please check your numbers.

3) In Figure 3, it would help the reader to visualize your point if you drew in the horizontal and vertical zero axes (and label zero rather than the arbitrary values that your software decides to label).

---

## [Author Response]

*1) The kinematic measurements obtained with eye coil recordings provide important support for the argument that BARMs represent a truly novel type of eye movement. More analysis and possibly eye coil data from more subjects are needed to strengthen this portion of the manuscript. Figure 5 reveals considerable variability across the three subjects recorded with eye coils […]*

We agree that our original sample of subjects studied with search coils was very small, if not marginal. This is why we followed the advice to add more subjects. Actually, we could meanwhile study two more subjects with search coils and consider the new data in the revised manuscript (see Figure 5 and Figure 7). The results contributed by these two new subjects are very similar to the ones based on the three subjects reported earlier. Similar to our first two subject Ha and Mo, the new two subjects show 2 components which are easily distinguishable. The amplitude of the first component amplitude does not have a relationship with its velocity (i.e. it does not exhibit a ‘main sequence’). On the other hand, the second component obeys a mean sequence relationship that has a slope lower than the one of the fast phase main sequence (see Figure 5 and Materials and methods). The patterns exhibited by the two new subjects with the ones exhibited by subjects Ha and Mo supports our interpretations that. Unlike the other subjects (old subjects Ha and Mo and the two new ones), exhibiting a clear separation of the two components, subject S´s pattern was characterized by a merger of the two components, compromising attempts to characterize the kinematics of specific components. We think that the addition of the new data, our original interpretation of the search coil recordings has become much more substantiated. In retrospect we are grateful to the reviewers for having put their finger on what was a weak spot in the argument.

[…] and much of the text describing the two kinetically distinct components of the blink-related eye movements seems to be describing results from the one subject shown in Figure 4 rather than a summary of results from the three subjects.

Now we are specifying which information are related to subject Ha and which are from all subjects. We added analyses from all subjects to the text.

Only one subject (S) seems to have been tested with both eye coil and video-oculography, and this subject shows atypical blink-related eye movement, making it difficult to compare the results obtained with the two methods.

It is true that subject S is the only one who participated in eye coil and video-oculography recording but the data from both methods are replicating the main result, which is resetting of the eye position after blinking. Even for subject S, one would not be able to compare the resetting between these two methods fairly since we had longer sessions (expected larger drifts) using the video eye tracking system compared with the coils. Including the subjects, who participated in the eye coil measurements, in video-oculgraphy would not add more information about the kinematic information that we collect from the coil.

However and to compare generally the eye position resetting associated with blinks, recorded by both methods, we added in Figure 7 the torsional resetting (recorded by the coils) which shows that it looks similar to the ones seen using the video-oculography as in Figure 3 and Figure 3—figure supplement 1.

*It is important to more clearly document which of the observations made with eye coils were consistent across subjects.*

We are documenting that more clearly in Results section, subsection “Blink-associated resetting of eye torsion is realized by a specific type of eye movement” now.

*2) There are several additional places where the statistics need to be clarified:*

*a) Although the statistics reveal no significant correlation between the torsional eye position just before the fast phase of optokinetic nystagmus and the amplitude of the quick phase (green in Figure 3 and Figure 3—figure supplement 1), inspection of the plots suggests that for a subset of the quick phases, there is a correlation with the eye position. The authors should comment on this.*

Obviously, the way we presented and explained the results in Figure 3 and Figure 3—figure supplement 1 were misleading. In fact the statistical analysis yields significant correlations between torsional shifts during fast phases and pre-shift eye positions in 3 out of 5 subjects. This is shown in Figure 3—figure supplement 1, and indicated by the red regression lines. We state explicitly in the legend now that red lines are only drawn if there was a significant correlation. We discuss that this correlation is inconsistent (only seen in 3 out of 5 subjects) and, even more relevant, the relationship is weak as indicated by the small correlation coefficients and very low slopes (from 0.03 to 0,22 vs. 0.41 to 0.9 for blinks). The clear quantitative differences, supporting the assumption of a qualitative distinction between BARMs and fast phases are now better recognizable in Figure 3 and Figure 3 which are based on a new analysis, a 2-D regression analysis, carried out as suggested by you (point 2e). These figures illustrate quantitatively the differential contribution of pre-shift eye position and pre-shift slow phase amplitude on torsional shift during blinks on the one hand and torsional shifts during fast phases on the other.

The impression that there might be a correlation, although our statistical analysis fails to reveal one is probably prompted by having a bimodal distribution of data points depending on the sign of torsional eye position: there is a positive torsional shift for negative eye positions and a negative torsional shift for positive eye positions. The sign of eye position and torsional shift during fast phases is a consequence of the direction of optic flow. If data from both direction are plotted – as done in Figure 3 – it appears as if there is a linear relationship although there is only a discrete distinction between fast phases for the two optic flow directions. Actually, if data from both directions were pooled a significant correlation resulted. To avoid this pitfall we analyzed data from each direction separately and plotted regression lines separately for each direction if the correlations were significant. For the analysis of blinks, however, we pooled the data because a direction-specific analysis did not reveal relevant differences in the offset and slope as it was seen for fast phases. This procedure is now described and justified in the Materials and methods section in more detail. The legends of Figure 3 and Figure 3—figure supplement 1 are adjusted and contain a referral to Materials and methods.

*b) Figure 3, the authors should clarify whether this is population or individual subject data, and why the data on fast phase saccades are fit with two lines, but there is a single correlation coefficient listed.*

The data are from the same subject as in 3A and we clarify that in the legend now.

We discussed the reason to plot two lines before (2a). For reasons of clarity, we list only one correlation coefficient per subject – the mean of the coefficients for the two directions. There is no reason to expect two qualitatively different processes for one or the other direction that would argue against taking the mean of both directions, which we did. This is mentioned now in the legends of Figure 3 and Figure 3—figure supplement 1 and in Materials and methods.

*c) Figure 3 shows the correlation coefficients with significance intervals estimated for each subject using a bootstrap analysis. A bootstrap analysis gives the standard error of the mean, and the customary use is to show the 95% confidence interval (the interval that includes 95% of the bootstrapped coefficient). Also, was the bootstrap analysis done with or without replacement?*

We changed the figure now to show the 95% confidence interval. We clarify in the text that bootstrapping was with replacement in the second paragraph of the Statistical Analysis section.

*d) Table 1 requires a description of the numbers given (what do the p-values compare? Do the numbers show means, medians, standard deviation, standard error, etc.).*

We changed the text such as to make it clear that the numbers given are means with standard deviations and that p values are reflecting the significance level of the correlation across all subjects.

*e) The authors show separate correlations between BARM amplitude and (1) pre-blink eye position, (2) pre-blink slow phase excursion. Wouldn't it be better to fit the BARM amplitude to a regression containing several terms (related to position, excursion and their interaction) and show the coefficients related to each term?*

We implemented your suggestion and we added this analysis (Figure 3). See also our comments regarding point 2a.

*f) The data for subjects A and S seem to be repeated in Figure 3 and Figure 6, but the numbers are slightly different in the two figures. The authors should comment on why this is the case.*

The numbers are slightly different because they are reflecting data collected in two different experiments the two subjects participated in. It is specified in the Materials and methods section, in the paragraph on subjects. For Figure 3—figure supplement 1, the data was collected from experiment 1 and for Figure 6—figure supplement 1 the data was collected from experiment 2, which was an additional control experiment for simulated blinks). Another difference is the amount of the data as experiment 1 included data collected from 8 days (16 blocks) for each subject in contrast with data collected in two days (4 blocks) for each subject in experiment 2.

[Editors' note: further revisions were requested prior to acceptance, as described below.]

*1) The group data for the result shown by the single example in Figure 6, which are currently shown in Figure 5—figure supplement A, C should be included in the main figure rather than a supplementary figure.*

We included Figure 6—figure supplement 1 in the main Figure 6.

*2) Please double check the values for the slopes in Figure 3—figure supplement 1. The numbers given for the slopes don't seem to correspond well with the slopes of the red regression lines, although this could be the result of fitting not one, but two regression lines to the data, and giving the mean of the two slopes. Even considering the latter, the relative slopes for the first and second panel from the left do not seem to correspond to my visual estimation of the relative slopes from the red lines – it may just be my eyes, but please check your numbers.*

You are right, the numbers are mixed in a way that the values of the slopes in Figure 3—figure supplement 1 are actually reflecting the correlation coefficients values (r) and what was presented correlation coefficient was actually the value of the slopes. Our apologies for this mistake, which is corrected now. This mistake motivated us to look at all figures and tables. They were all correct except for Table 1 which had the same problem (in the first three rows). We corrected this table too.

*3) In Figure 3, it would help the reader to visualize your point if you drew in the horizontal and vertical zero axes (and label zero rather than the arbitrary values that your software decides to label).*

We drew in horizontal and vertical zero axes to improve the visualization. Moreover, changed the axes labels in a way that it includes zeros now.

PS: There is no change on the main text except the legend of Figure 6 and Figure 6—figure supplement 1.